# Tailored PGE2 Immunomodulation of moDCs by Nano-Encapsulated EP2/EP4 Antagonists

**DOI:** 10.3390/ijms24021392

**Published:** 2023-01-11

**Authors:** Johanna Bödder, Leanne M. Kok, Jonathan A. Fauerbach, Georgina Flórez-Grau, I. Jolanda M. de Vries

**Affiliations:** 1Department of Tumor Immunology, Radboud Institute for Molecular Life Sciences, Radboud University Medical Center, 6525 GA Nijmegen, The Netherlands; 2R&D Reagents, Chemical Biology Department, Miltenyi Biotec B.V. & Co. KG, 51429 Bergisch Gladbach, Germany

**Keywords:** prostaglandin E2, EP2/EP4 antagonist, dendritic cells, nanoparticles, delivery

## Abstract

Prostaglandin E2 (PGE2) is an important maturation mediator for dendritic cells (DCs). However, increased PGE2 levels in the tumor exert immunosuppressive effects on DCs by signaling through two E-Prostanoid (EP) receptors: EP2 and EP4. Blocking EP-receptor signaling of PGE2 with antagonists is currently being investigated for clinical applications to enhance anti-tumor immunity. In this study, we investigated a new delivery approach by encapsulating EP2/EP4 antagonists in polymeric nanoparticles. The nanoparticles were characterized for size, antagonist loading, and release. The efficacy of the encapsulated antagonists to block PGE2 signaling was analyzed using monocyte-derived DCs (moDCs). The obtained nanoparticles were sized between 210 and 260 nm. The encapsulation efficacy of the EP2/EP4 antagonists was 20% and 17%, respectively, and was further increased with the co-encapsulation of both antagonists. The treatment of moDCs with co-encapsulation EP2/EP4 antagonists prevented PGE2-induced co-stimulatory marker expression. Even though both antagonists showed a burst release within 15 min at 37 °C, the nanoparticles executed the immunomodulatory effects on moDCs. In summary, we demonstrate the functionality of EP2/EP4 antagonist-loaded nanoparticles to overcome PGE2 modulation of moDCs.

## 1. Introduction

Dendritic cells (DCs) are crucial for antigen-specific immune response induction. The vaccination of cancer patients with DCs has been beneficial in some patients and is considered safe [1]. Until recently, most studies have been conducted employing monocyte-derived DCs (moDCs), a type of DC generated in vitro, due to well-known protocols [2,3]. For vaccine efficacy, moDCs maturation is crucial for immune response induction and is ensured by using a cytokine cocktail containing tumor necrosis factor (TNF) α, Interleukin (IL-) 6, IL-1β, and Prostaglandin E2 (PGE2) [4,5,6]. In particular, PGE2 has dichotomous functions in the immune system and is described as a pro-inflammatory mediator with immunosuppressive activities [7]. PGE2 is a lipid metabolite derived from arachidonic acid via the enzymes cyclooxygenase (COX) 1 and 2. The four different EP receptors (EP1-EP4) of PGE2 are expressed on multiple cell types in the human body, activate different pathways, and have a distinct affinity for PGE2 [7,8]. PGE2 attracts and activates immune cells thereby supporting local inflammation [7]. Cancer is often associated with inflammation and this might be partly due to the overexpression of COX-2 and PGE2 in various cancers [9,10]. PGE2 suppresses antitumoral immunity by inhibiting effector immune cells such as T- and NK cells and leading to the differentiation of suppressive immune cells [11,12]. In addition, PGE2 has opposing effects on DCs, mostly shown for mouse DCs and human moDCs. PGE2 signals through two receptors, EP2 and EP4, on human DCs which might be the reason for the reported diverse effects [13,14,15,16,17]. On the inflammatory side, PGE2-matured moDCs show a concentration-dependent increase in co-stimulatory marker expression (CD86, CD83, and HLA-DR), higher T cell proliferation, and upregulation of CCR7 expression which enhances the migration of moDCs [6,14,15,18]. Nevertheless, PGE2 also promotes local and systemic DC dysfunction in cancer [19,20,21,22].

Overall, tumor-derived PGE2 has immunosuppressive effects on immune cells, thus it is hypothesized that preventing these immune-inhibiting effects could increase the efficacy of cancer immunotherapy [21,23].

In the first phase I clinical trial in which the PGE2-EP4 pathway is manipulated, oral administration of EP4 antagonist demonstrated immunomodulatory effects and manageable tolerability in cancer patients [24]. Blocking both EP2 and EP4 receptors could enhance the efficacy and is currently being investigated in another clinical trial in combination with anti-PD-1 antibody therapy (NCT04344795). However, the systemic administration of EP antagonists can impact multiple cell types; therefore, a more specific drug delivery approach might reduce side effects and toxicity in patients [25]. Nanoparticles (NPs) have emerged as a unique delivery tool in cancer immunotherapy due to their various beneficial properties such as sustained release, cargo protection from degradation, and specific cell-targeted delivery [26,27]. NPs can be synthesized from a wide range of materials, for example, biodegradable and biocompatible polymers. One widely used and FDA-approved polymer is poly (D, L-lactic-co-glycolic acid) (PLGA) [28].

We rationalized that providing EP2 and EP4 antagonists encapsulated in PLGA NPs to patients could further reduce PGE2 immunosuppressive effects. The simultaneous delivery of both synergistic antagonists in a controlled quantity to the same target cells could avoid random dose delivery and the side effects of systemically blocking PGE2. In this study, we encapsulate EP2 (AH6809) (aEP2) and EP4 (L-161,982) (aEP4) antagonists in PLGA NPs for the first time and therefore we provide the initial step towards a new delivery approach.

## 2. Results

### 2.1. EP2 and EP4 Antagonists Are Successfully Encapsulated in PLGA Nanoparticles

In this study, we encapsulated antagonists for EP2 (aEP2) and EP4 (aEP4) alone or in combination with PLGA NPs. In total, five different NP formulations were produced: NP without antagonist (NP (empty)), NPs with loading 2 weight percent (wt%) aEP2 (NP (aEP2)), or 2 wt% aEP4 (NP (aEP4)), a combination of 3 wt% aEP2 and 1 wt% aEP4 (NP1 (aEP2 + aEP4)), and with 3 wt% aEP2 and 2 wt% aEP4 (NP2 (aEP2 + aEP4)). Different batches of each NP formulation were characterized for size, polydispersity index (PDI), wt% of encapsulated antagonists, and encapsulation efficacy (EE%) (Table 1).

The mean particle size was between 211 nm for NP (empty) and 261 nm for NP1 (aEP2 + aEP4). The PDI ranged from 0.06 for NP (aEP2) to 0.13 for NP1 (aEP2 + aEP4) indicating a monodisperse size distribution. The quantity of encapsulated antagonists was assessed with high-performance liquid chromatography (HPLC) from which the wt% encapsulated and EE% were calculated. The aEP2 was detected at a retention time of 13 min at 244 nm and aEP4 after 16 min at 249 nm (Appendix A).

For NPs separately loaded with 2 wt% aEP2 and aEP4, 1.24 and 0.63 wt% were encapsulated, respectively. The combination of both antagonists in the same NP increased the encapsulation of aEP2 to 3.12 wt% while aEP4 displays a similar encapsulated wt% for all NP formulations. Consequently, the EE% of aEP2 increased from 21% for the NP (aEP2) to 67% for the NP1 (aEP2 + aEP4) while the EE% of aEP4 varied from 14% for NP2 (aEP2 + aEP4) to 25% for NP1 (aEP2 + aEP4). The high standard deviations indicate variations between the various batches of prepared NPs.

In general, the PGE2 receptor antagonists are successfully encapsulated in PLGA NPs with a higher EE% for aEP2 especially when both antagonists are encapsulated simultaneously.

### 2.2. EP2 and EP4 Antagonists Display High Initial Release

To determine the release profile of the antagonists, the NPs were incubated at 37 °C and 4 °C, and the released antagonists were measured with HPLC at different timepoints.

We detected a high initial release of both antagonists from all NP formulations (Appendix A). Table 2 displays the remaining percentage of aEP2 and aEP4 in the NPs after the first 15 min at 37 °C. The NPs with separately encapsulated antagonists contained 0.7% aEP2 and 17.4% aEP4. However, the NP with the combination of the antagonists encapsulated retained higher percentages of antagonists with 31.1% and 32.6% aEP2, and 44.0% and 36.9% aEP4 in the NP1 and NP2, respectively.

The high short-term release of both antagonists at 37 °C is followed by a plateau, indicating no or minor further release after the first 15 min of incubation. The release of aEP4 seems to depend on the temperature indicated by a low release of approximately 30% at 4 °C which is increased to 60% at 37 °C for the combination NPs. The release of aEP2 is higher compared to aEP4 and independent of the temperature (Appendix A).

Based on the release profiles it seems that approximately 30% aEP2 and 40% aEP4 are stably encapsulated in the combination NPs. Overall, these findings indicate that the dual encapsulation of aEP2 and aEP4 results in robust encapsulation.

### 2.3. Encapsulated EP2/EP4 Antagonists Modulate MC + PGE2-Induced moDC Phenotype

The potential of the encapsulated antagonists to influence the PGE2-mediated phenotype of moDCs was assessed. MoDCs were stimulated with a PGE2-containing DC-maturation cocktail (MC) in the presence of different NP formulations followed by measuring co-stimulatory marker expression. The MC, containing IL-6, IL-1β, and TNFα increased expression of co-stimulatory markers (CD83, CD40, HLA-DR, and CD86) indicating maturation of moDCs compared to the untreated (NT) condition (Figure 1). The addition of PGE2 to the MC (MC + PGE2) resulted in increased expression of CD83 and CD86 on moDCs.

To facilitate the comparison of different NP formulations on moDC maturation, NPs were added in a final concentration of 60 µM of aEP2. For samples containing a combination of antagonists, the corresponding aEP4 concentration was calculated per NP batch and the NP dose was adjusted to an aEP4 concentration between 2.5 and 5 µM. The addition of NP (aEP2) significantly downregulated CD40, HLA-DR, and CD86 expression on moDCs during MC + PGE2 stimulation. While NP (aEP4) resulted in a slightly diminished expression of co-stimulatory markers. However, the combination of both antagonists in the same NP showed a synergistic effect and a significant decrease in the expression of CD83 (*p* = 0.026) and CD86 (*p* = 0.035).

The addition of empty NP did not affect marker expression by matured moDCs and CD40 and CD86 expression significantly decreases with encapsulated antagonists compared to empty NPs treated moDCs, indicating that the encapsulated aEP2 and aEP4 specifically blocked PGE2 signaling. The PLGA NPs were well tolerated as no changes in moDC viability were measured.

Overall, encapsulated aEP2 and aEP4 modulated the MC + PGE2-induced maturation phenotype of moDCs. The combination of both antagonists in one NP displayed the lowest antagonist release and was demonstrated to be more potent in blocking PGE2 and therefore tested in further experiments.

### 2.4. Initial Antagonist Release from Encapsulated aEP2/aEP4 Does Not Affect Blocking Capacity

To assess whether the initial release of antagonists from the NP is affecting the capacity to modulate the marker expression of MC + PGE2-matured moDCs, NPs were pre-incubated in PBS at 37 °C for 15 min, washed, and resuspended in fresh PBS at the same concentration.

As shown in Figure 2, both PBS-incubated and non-incubated NP2 (aEP2 + aEP4) decreased MC + PGE2-mediated expression of CD40, HLA-DR, and CD86. In comparison to control NP (empty), both PBS-incubated and non-incubated NP2 (aEP2 + aEP4) significantly decreased CD83 and CD86 expression. Only the CD83 expression on MC + PGE2 moDCs was significantly lower in the presence of non-incubated NP2 (aEP2 + aEP4) compared to the PBS-incubated NPs (*p* = 0.01) (Figure 2). Furthermore, a lower concentration of encapsulated antagonists (30 µM aEP2), either PBS-incubated or non-incubated NP2 (aEP2 + aEP4) decreased HLA-DR, CD40, as well as CD86 expression on MC + PGE2-matured moDCs (Appendix A). In addition, we determined whether the supernatant of the PBS-incubated NPs could modify PGE2-induced co-stimulatory marker expression on moDCs. Therefore, supernatant of the PBS-incubated NPs and resuspended PBS-incubated NPs were added to MC + PGE2-matured moDCs. The supernatant of the PBS-incubated NPs had a significant impact on CD83 expression indicating the antagonists released from the NPs act as soluble antagonists and prevent PGE2-induced CD83 expression (Appendix A).

In summary, NPs encapsulating aEP2 and aEP4 are capable, even after the initial high release of antagonists, to counteract the effect of PGE2 on MC + PGE2 matured moDC.

### 2.5. The Immunomodulatory Effects of Encapsulated EP2/EP4 Antagonists Are Time-Course Stimulation Independent

In the following, we analyzed whether the continuous presence of NP is necessary for the modulation of MC + PGE2-induced moDCs phenotype. One hour after NP (aEP2 + aEP4) addition, moDCs were washed and the MC + PGE2 was added for 24 h. The washed moDCs illustrate a lower marker expression after NP (aEP2 + aEP4) stimulation compared to the MC + PGE2 control, although not significant (Figure 3). It seems that NP2 (aEP2 + aEP4) results in a further decrease in marker expression compared to NP1 (aEP2 + aEP4), especially in the higher antagonist concentration (Figure 3 and Appendix A). As seen before there is a concentration-dependent trend of antagonist concentration (Appendix A).

In addition, we examined the impact of the extended presence of NP prior to adding MC + PGE2 on the moDCs phenotype. Over four hours CD83, CD40, and CD86 expression was significantly decreased after NP2 (aEP2 + aEP4) stimulation compared to the MC + PGE2 control (Figure 4 and Appendix A). When analyzing the marker expression after NP stimulation from the first timepoint to the later one, there is no significant decline. Nevertheless, compared to the MC + PGE2 control after one hour stimulation CD83 expression is significantly decreased after NP2 (aEP2 + aEP4) for all timepoints. CD40 and CD86 expressions show a similar trend (Appendix A).

Together, these results provide important insights into the efficacy of the encapsulated EP antagonists to prevent PGE2-mediated upregulation of co-stimulatory marker expression on moDCs. It can be concluded that the NPs (aEP2 + aEP4) execute their immunomodulatory effect on moDCs within the first hour of stimulation. Since neither the prolonged stimulation with NPs nor washing the cells results in differences in co-stimulatory marker expression.

### 2.6. Encapsulated EP2/EP4 Antagonists Show Strongest Modulatory Effect with Lowest PGE2 Concentration

PGE2 has the unique capacity to upregulate chemokine receptor CCR7, detrimental for DC to migrate towards its ligands CCL19 and CCL21 present in lymph nodes where they can interact with naïve T cells. We investigated whether the encapsulated antagonists could modulate this PGE2-mediated CCR7 expression on moDCs.

The percentage of CCR7^+^ moDCs increased significantly with 1 µM (*p* = 0.029) and 28 µM (*p* = 0.001) of PGE2 compared to 0.2 µM (Figure 5). The encapsulated antagonists reduced CD83 and CCR7 expression when stimulated with 0.2 µM PGE2 but the antagonistic effect is dampened with 1 or 28 µM PGE2.

CD83 expression is upregulated with the higher PGE2 concentrations significantly for 28 µM PGE2 (*p* = 0.0255) and the CCR7 expression follows a similar trend (Appendix A). Higher PGE2 concentrations did not affect CD40 and CD86 expression (Appendix A). For all markers, the addition of encapsulated NP (aEP2 + aEP4) results in a slightly lower expression compared to the addition of soluble antagonists.

Overall, the NPs show the strongest modulatory effect on the moDCs phenotypes with the physiologically relevant 0.2 µM PGE2 concentration.

## 3. Discussion

Here, we show that PLGA NP-encapsulated EP2 and EP4 antagonists can overcome PGE2 modulation of moDCs. Polymeric nanoparticle-based delivery systems have emerged in the field of cancer treatment [27,29]. NPs composed of biodegradable and biocompatible polymers possess advantages for drug delivery such as increased drug stability, sustained release, or specific cell targeting [30]. PLGA NPs have been previously used for the encapsulation of chemotherapeutic reagents or tumor antigen adjuvant combinations to induce tumor-specific immune responses [29,31]. We are currently determining the safety of tumor antigen and adjuvant-loaded PLGA NPs in a phase I clinical trial NCT04751786. Another approach is the encapsulation of immunomodulatory drugs to target and reverse immunosuppressive effects in the tumor microenvironment (TME) [26,27,28]. PGE2 is an immunosuppressive mediator excessively present in cancer patients [9,10]. In recent decades, efforts were undertaken to inhibit PGE2 signaling in cancer patients, but they were associated with severe side effects [32,33,34]. However, blocking the receptors of PGE2 with specific antagonists shows promising results [23,24].

NP characteristics, such as size and drug loading, are crucial for the efficacy of drug delivery. Prior studies have noted the importance of NPs size for their uptake by cells. DCs are superior in the uptake of smaller NPs not exceeding 500 nm, larger particles are preferably taken up by macrophages [35,36,37]. The PLGA NPs produced in this study have an optimal size for DC uptake, 210 to 260 nm, and a small PDI (<0.13) indicating a uniform size distribution [38]. The NP size varies per NP formulation with empty NPs displaying the smallest size and sizes increased for NPs with the combination of antagonists encapsulated. A possible explanation is that NP size increases with cargo loading, as described before [39,40,41]. The small standard deviation for the size measurement confirms that the size difference is existent and is due to cargo loading.

We observed differences in the wt% encapsulated and EE% between the NPs encapsulated separately or in combination. Within the separately encapsulated antagonists, NP (aEP2) displays a higher wt% encapsulated compared to NP (aEP4). These data must be interpreted with caution due to batch differences. Nevertheless, while aEP4 is similarly encapsulated for the various NP formulations, aEP2 encapsulation increased when co-encapsulated with aEP4. Although modifications in the NP synthesis methods could potentially increase EE% [30,42,43], we decided to not further optimize the loading of individual antagonists. The NPs with aEP2 and aEP4 encapsulated show a high EE% and targeting EP2 and EP4 receptors simultaneously is suggested to be more effective on a functional level [21].

In addition, the EP4 receptor has a higher affinity to PGE2 than EP2 [44]. Thus, to block EP2-PGE2 signaling an increased aEP2 concentration is necessary. For this reason, we loaded a higher wt% of aEP2 than aEP4 into the NPs. The increased wt% of aEP2 in the NPs (aEP2 + aEP4) is convenient for the controlled cellular delivery of the antagonists.

Besides drug loading, the release rate of the encapsulated drug is an important parameter for the characterization of NP drug delivery. We determined the short-term release within the first four hours since the NPs should be taken up by DCs shortly after administration [45,46,47]. Overall, all NP formulations displayed a high initial burst release. The NPs with separate aEP2 encapsulated exhibited the fastest and strongest release. There are several possible explanations for this result. The NPs (aEP2) were the smallest NPs produced and others revealed that smaller NPs display a faster antigen release [30]. Moreover, aEP2 has higher aqueous solubility, indicated by the earlier retention time during HPLC measurement. This could result in an easier diffusion out of the NPs for aEP2 compared to aEP4 [42].

For the combination NPs, the initial release for both antagonists ranges from 60% to 70%. The high initial burst release of cargo from PLGA NPs has been described before [48,49]. There are two potential reasons for antigen release from PLGA NPs. Either the antigens escape from pores in the polymer depending on the degradation and porosity of the polymer or they are released from the surface layer of the NPs. The latter is associated with the efficacy of antigen loading and is likely the reason for our high release [30,50]. The high release was detected directly upon suspension (t = 0) indicating the release of poorly entrapped or loosely attached antagonists from the surface of the NPs. These results imply that the initial encapsulation efficacies should be interpreted with caution. We attempted to minimize the antagonist release for cellular assays by preparing the NPs directly before adding them to the cells with 4 °C cold PBS but only aEP4 exhibited a lower release at 4 °C. Regarding the cellular assays these results imply that around 60% of the antagonists are provided in a soluble form when the NPs are not previously washed with PBS. Although the high release is not ideal, the soluble antagonists have been evaluated extensively and are considered safe [24]. Moreover, after the first 15 min, there is no further release indicating the stable encapsulation of the remaining antagonists in the NPs within the first four hours. Hence, we analyzed the functionality of the encapsulated antagonists in various in vitro cell assays.

PGE2 together with pro-inflammatory cytokines (TNFα or IL-1ß) matures DCs and upregulates the expression of markers such as CD80, CD86, CD83, and HLA-DR via EP2 and EP4 receptor signaling [6,17,51]. This is in accordance with our results, elevated expression of especially CD83 on moDCs was observed by adding PGE2 to the maturation cocktail consisting of TNFα, IL-1β, and IL-6. Encapsulated aEP2 and aEP4 successfully prevented PGE2-induced CD83 expression demonstrating the functionality of the NPs. The specificity of the encapsulated antagonists to block PGE2 signaling is indicated by the lack of modulatory effects with empty NPs. This result is supported by a concentration-dependent downregulation of marker expression of the antagonists.

Comparing the different NP formulations, blocking of the EP2 receptor led to a stronger phenotypic modulation than EP4. Suggesting that PGE2 signals mainly through EP4 receptors leading to the mature phenotype of moDCs. Nevertheless, when blocking EP2 and EP4 receptors simultaneously the phenotypical changes were more distinct than with the separate receptor targeting.

Upon further testing, we determined that despite the high initial antagonist release the NP (aEP2 + aEP4) retained sufficient antagonists to remain functional. Furthermore, exposure of moDCs to NP (aEP2 + aEP4) for only one hour was enough to observe marker downregulation. Together these results are promising and illustrate the effectiveness of the encapsulated antagonists. However, based on the results of this study we can only speculate whether the NPs execute their function via sustained release or require cellular uptake. We observed that the encapsulated antagonists display a stronger downregulation of PGE2 + MC-induced marker expression compared to the soluble antagonists, indicating an additional effect of the NPs. The effectiveness of the washed NPs strengthens the idea that the NPs at least partially function via NP uptake.

We employed the maturation of moDCs with PGE2 as a model system to show the functionality of encapsulated antagonists. A relatively low concentration of PGE2 (0.2 µM) was used in comparison to previously reported concentrations applied for ex vivo maturation of moDCs for clinical applications (28 µM) [52]. This high concentration ensures the induction of a migratory phenotype via the upregulation of CCR7 by PGE2 [14,15,18]. Consistent with the literature, CCR7 expression on moDCs also here increased with higher PGE2 concentrations. While CCR7 and CD83 expressions were PGE2-concentration-dependent, other co-stimulatory markers were not affected by an increased PGE2 concentration. Neither the soluble nor the encapsulated antagonists influenced CCR7 or CD83 expression with 28 µM PGE2, most likely due to the higher affinity of PGE2 for the receptors compared to the antagonists. Previously, the presence of 0.2 µM of PGE2 in vivo is described as a physiologically relevant PGE2 concentration [23]. At this concentration, encapsulated antagonists NP (aEP2 + aEP4) display immunomodulatory effects.

Overall, we showed the feasibility to encapsulate aEP2 and aEP4 in PLGA NPs with an ideal NP size for DCs uptake. The NPs with the combined antagonists with the higher EE% and lower release were the most efficient to modulate PGE2 induced phenotype with a clinically relevant PGE2 concentration. PGE2 in the tumor microenvironment and has various immunosuppressive effects on immune cells including DC function and development [20,21,22,53,54]. Blocking both EP2 and EP4 receptors to diminish these immunosuppressive effects is currently being investigated in a clinical trial (NCT04344795). Future research should explore whether the encapsulation of aEP2 and aEP4 in PLGA NPs could improve efficacy by dose-controlled delivery of both antagonists to the same target cell. First, the effect of the NP (aEP2 + aEP4) on different immune cells including primary DCs should be assessed. This will provide a rationale for the here-described PLGA NP as a delivery tool for EP2 + EP4 antagonists for cancer patients.

## 4. Materials and Methods

### 4.1. Nanoparticle Synthesis

PLGA NPs were synthesized by the double emulsion solvent evaporation method. Briefly, 100 mg of PLGA copolymers Resomer 50:50 Resomer RG 502H (Evonik Industries, Darmstadt, Germany) was dissolved in 3 mL of dichloromethane (DCM) (Merck, Darmstadt, Germany) as an organic solvent. The EP antagonists EP2: AH6809 (Cayman Chemical, Ann Arbor, MI, USA) and EP4: L-161982 (Cayman Chemical) were dissolved in Dimethyl sulfoxide (DMSO) (Merck) at 9 mg/mL (30 mM) and 13.5 mg/mL (20 mM), respectively. Depending on the NP batch, 1 to 4 wt% of the antagonists was added alone or in combination with the PLGA in DCM. The PLGA and EP antagonists were sonicated (Branson Ultrasonics, Brookfield, CT, USA) for 30 s with a 10% amplitude. Next, this emulsion was drop-wise added to 25 mL of 2.5% Poly-vinyl alcohol (PVA) (Sigma Aldrich Chemicals, Saint Louis, MO, USA) under sonication (2 times 58 s 20% amplitude). Upon overnight evaporation of the solvent at 4 °C, the NPs were collected and centrifuged at 11,000 rpm for 20 min at 4 °C. Subsequently, NPs were washed with Milli-Q (11,000 rpm for 20 min at 4 °C) and resuspended by sonication. After three washing cycles, the NPs were dissolved in Milli-Q, frozen at −80 °C followed by lyophilization, and stored at −20 °C.

### 4.2. Nanoparticle Characterization

The NPs were characterized for average size and PDI via dynamic light scattering (DLS). Therefore, after lyophilization, a solution of 0.5 mg/mL of NPs in Milli-Q was measured with a Zetasizer (Nanotrace Flex, Microtrac Montgomeryville, PA, USA) and the Microtrac FLEX Software version 11.1.0.2 (Microtrac, Inc, Montgomeryville, PA, USA). The antagonist encapsulation was determined using a reverse-phase (RP-HPLC) (Shimadzu, Kyoto, Japan). A sample of 10 mg/mL NPs was dissolved in DMSO and further diluted in acetonitrile (ACN) and water for injection (WFI) (1:1) with 0.01% trifluoroacetic acid (TFA) to 5 mg/mL. The dissolved NPs were spun down at 10,000 rpm for 10 min at 4 °C and 25 µL of the supernatant was injected into the HPLC. For the HPLC analysis and an X select, C 18 CSH XP column (4.6 mm × 100 mm, 3.5 µm particle size, 175Å pore size) (Waters, Milford, MA, USA) was used. The flow rate was 1 mL/min with a linear gradient from 5% to 90% ACN + 0.01% TFA in WFI + 0.01% TFA. The absorbance of aEP2 was detected at 244 nm, and aEP4 at 249 nm with the LabSolutions program version 5.99 (Shimadzu Kyoto, Japan). The amount of antagonists encapsulated was calculated based on a linear standard curve. The wt% encapsulated was calculated with Equation (1). The EE% was calculated via Equation (2).
(1)$$weight\;percent\;encapsulated\;\left(\%\right)=\frac{weight\;of\;measured\;antagonist}{weight\;of\;NPs}*100$$
(2)$$encapsulation\;efficacy\;\left(\%\right)=\frac{wt\%\;measured}{wt\%\;added\;for\;synthesis}*100$$

### 4.3. Release Assay

The release of the EP antagonists over time was measured at 4 °C and 37 °C. A suspension of NPs in PBS was stored at 4 °C and 37 °C. At different time points (t = 0, 15 min, 30 min, 2 h, and 4 h), two samples of an equal volume of 100 µL were taken from the NP suspension. One sample (“total”) was diluted directly 1:1 with DMSO while a second sample (“pellet”) was spun down at 10000 rpm for 10 min at 4 °C. From the “pellet” sample 50 µL of the supernatant was taken and diluted with DMSO 1:1 and labeled as “supernatant” sample. The rest of the supernatant was discarded, and the pellet was dissolved in 100 µL DMSO. All samples were further diluted with ACN + WFI (1:1) with 0.01% TFA centrifuged at 10000 rpm for 10 min at 4 °C and 25 µL were injected into the HPLC to measure the aEP2 and aEP4 content as described before. The percentage of released antagonists was calculated according to the amount of antagonists remaining in the pellet compared to the total amount at each timepoint. 

### 4.4. Peripheral Blood Mononuclear Cell Isolation

Peripheral blood mononuclear cells (PBMCs) were isolated from buffy coats of healthy donors (Bloodbank Sanquin, Nijmegen, Netherlands) by Lymphoprep (VWR, Radnor, PA, USA) density gradient centrifugation. In short, buffy coats were diluted to 180 mL with RT diluting buffer (PBS + 2 mM EDTA (UltraPure™ 0.5M EDTA, pH 8.0 Invitrogen, Waltham, MA, USA)) and divided into 6 × 50 mL tubes. After carefully pipetting 10 mL of Lymphoprep underneath, the tubes were centrifuged at 20 min room temperature (RT) 2100 rpm without a break. The layer of PBMCs was collected in 50 mL tubes and washed with diluting buffer (10 min, RT, 1800 rpm). The cell pellets were pooled and washed three times with cold washing buffer (PBS + 2 mM EDTA + bovine serum albumin (Roche, Bazel, Zwitserland)) (5 min, 4 °C, 1500 rpm). To lyse erythrocytes, pellets were resuspended in 5 mL Ammonium-chloride-potassium (ACK) lysis buffer (50 mM NH4Cl, 10 mM KHCO3, 0.1 mM Na2EDTA, pH = 7.2–7.4) for 5 min at RT. The cells were washed with wash buffer for 5 min, at 4 °C, 1500 rpm, resuspended in 50 mL wash buffer, and counted. 

### 4.5. Isolation of CD14+ Monocytes and Differentiation of Monocyte-Derived DCs (moDCs)

MoDCs were differentiated from CD14^+^ monocytes isolated from PBMCs. CD14^+^ monocytes were isolated using the MACS CD14^+^ isolation kit (130-050-201, Miltenyi Biotec, Bergisch Gladbach, Germany) following the manufacturer’s protocol. Per 1 × 10^7^ cells, 80 µL of wash buffer and 20 µL of CD14^+^ beads were added to the cells. After 15 min of incubation at 4 °C, the PBMCs were washed with 50 mL of washing buffer (5 min, 4 °C, 1500 rpm). The pellet was resuspended in 0.5 mL per 1 × 10^6^ cells. Then, 0.5 mL of the cell suspension was added to LS columns (130-042-901, Miltenyi Biotec), which were pre-washed with washing buffer (2 mL). After the flowthrough was collected the columns were washed three times with 1 mL of washing buffer. The columns were taken off the magnet and the CD14^+^ cells were flushed out with 2 mL of washing buffer and counted. The CD14^+^ cells were resuspended in 0.5 million cells/mL in X-VIVO medium (Lonza, Bazel, Zwitserland) + 2% human serum (Sanquin), IL-4 (300 IU/mL, 130-093-924, Miltenyi Biotec) and GM-CSF (450 U/µL, 130-093-868, Miltenyi Biotec). To obtain immature moDCs, CD14^+^ cells were cultured for six days in 96-well plates (100,000 cells in 200 µL per well). After three days, cytokines IL-4 (300 IU/mL) and GM-CSF (450 U/µL) were again added. 

### 4.6. In Vitro Stimulation of moDCs with EP2 and EP4 Antagonists

After six days, moDCs were stimulated with encapsulated aEP2 and aEP4. The NPs were dissolved in PBS to a final concentration of 30 µM or 60 µM of aEP2 and between 1.25 µM and 5 µM of aEP4. For NPs with the combination of antagonists, the concentration of NPs was adjusted for the aEP2 concentration. The according concentration of aEP4 was calculated and the NPs with solely aEP4 were adjusted to the same concentration. The encapsulated antagonists were added to the cells in 25 µL of PBS (10% of the final volume). NPs without antagonist (NP (empty)) matched to the highest NP concentration were included as control. After 2 h of NP stimulation, 25 µL of maturation cocktail (MC) consisting of IL-1β (5 µg/mL 130-093-898 Miltenyi Biotec), IL-6 (15 µg/mL 130-093-933 Miltenyi Biotec), and TNFα (10 µg/mL, 130-094-014 Miltenyi Biotec) was added with or without the addition of PGE2 (0.2 µM, Pfizer, New York City, NY, USA). For indicated experiments, the PGE2 concentration was increased to 1 and 28 µM. As control, moDCs were left untreated (NT) and the final volume was adjusted with 25 µL of X-VIVO medium and 25 µL of PBS. Each condition was plated and analyzed in duplicate. 

To determine whether the NPs still have an effect after the burst release of the antagonists, the dissolved NPs were incubated for 15 min at 37 °C. The NPs were centrifuged, and the supernatant was harvested. Subsequently, NPs were dissolved in PBS to the same concentration as described above. Then, 25 µL of the PBS-incubated NPs and of the non-incubated NPs with the same antagonist concentration or supernatant of washed NPs were added to the cells. 

To investigate the effect of washing the cells after the incubation with the antagonist, plates were centrifuged (2 min 1500 rpm) after 1 h of incubation with the antagonists. The supernatant was discarded and 200 µL of X-VIVO medium was added to each well. After 1 h of incubation, MC + PGE2 was added.

To analyze the effect of the duration of the antagonist stimulation, the MC + PGE2 was given 1, 2, 3, or 4 h after the addition of the antagonists to the moDCs. After 24 h of incubation at 37 °C, the moDCs were harvested for staining and assessment using flow cytometry. 

To determine the phenotype, moDCs were stained in 25 µL in 96 well plates. Live/dead staining was performed with e Fluor450 viability dye (1:2000, 65-0863-14 ThermoFisher, Waltham, MA, USA) in PBS for 20 min at 4 °C in the dark. The cells were washed with PBA (PBS + 5% FBS + 0.01% NaH3) and stained for 20 min at 4 °C with monoclonal antibodies binding to CD86 (1:30, PE-Cy7, 2331 (FUN-1), 561128, BD, Franklin Lakes, NJ, USA), CD83 (1:30, FITC, HB15e, 556910 BD), CD40 (1:30, PE, MAB89, IM1936U, Beckman Coulter, Brea, CA, USA), HLA-DR (1:30, PerCP, L243, 307628, BioLegend, San Diego, CA, USA), and CD197 (CCR7) (1:200, APC, REA546, 130-120-466, Miltenyi Biotec). The moDCs were washed twice with PBA before being resuspended in PBA. Fluorescent intensities were measured on a FACS Verse (BD). The data were analyzed with the FlowJo software version 10 (BD, Franklin Lakes, NJ, USA).

### 4.7. Statistical Analysis

The significance of differences between different DC culture conditions was analyzed by one-way analysis of variance (ANOVA) with Tukey multiple comparison corrections using GraphPad Prism 8 (Version 8.0.2. GraphPad Software San Diego, CA, USA). All culture conditions were compared to every other condition. The *p* values ≤ 0.05 were considered statistically significant.

## Figures and Tables

**Figure 1 ijms-24-01392-f001:**
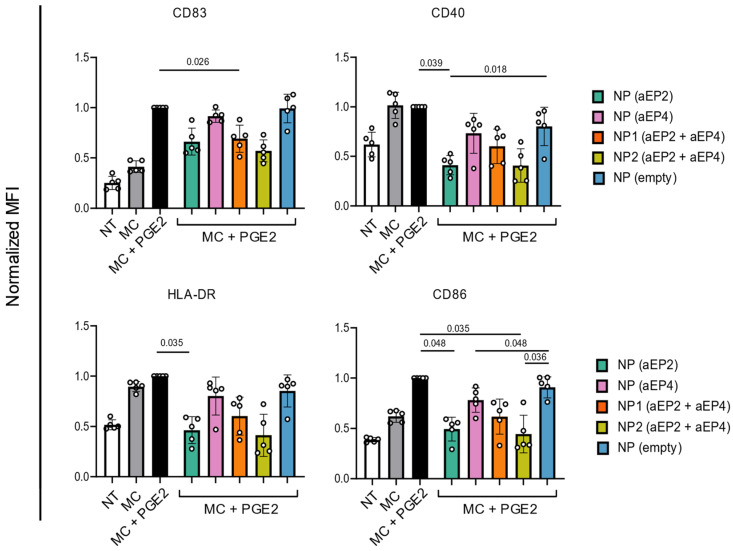
Normalized expression of CD83, CD40, HLA-DR, and CD86 on moDCs after 24 h in the presence or absence of NP formulations. Control moDCs were left untreated (NT, white bar), stimulated with a maturation cocktail (MC, gray bar), or MC with 0.2 µM PGE2 (MC + PGE2, black bar). Test condition moDCs were stimulated with MC + PGE2 and NPs with either aEP2 or aEP4 alone or in combination (green, pink, orange, and yellow bars, respectively). The NPs were added in a final concentration of 60 µM aEP2 and the aEP4 concentration was adjusted to the according concentration calculated in the combination NPs. Empty NPs were used as control (blue bar). Expression was normalized to the expression of the marker on moDCs stimulated with MC + PGE2. Each dot represents one donor and the bar is the mean ± SD for n = 5. *p* values were calculated on non-normalized data with one-way ANOVA with Tukey multiple comparison correction.

**Figure 2 ijms-24-01392-f002:**
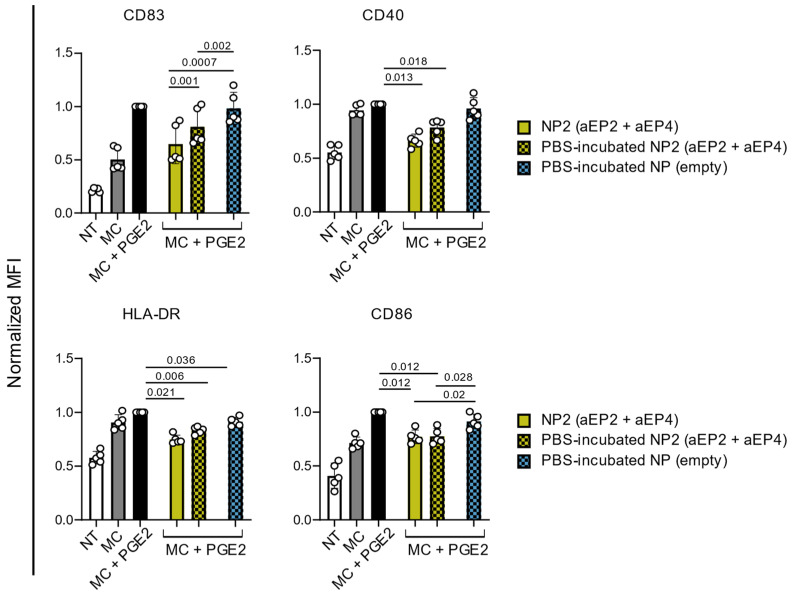
Normalized expression of CD83, CD40, HLA-DR, and CD86 on moDCs after 24 h of stimulation. Control moDCs were left NT or stimulated with MC either in the presence or absence of PGE2. MoDCs were stimulated with MC + PGE2 with either NPs PBS-incubated or non-incubated NPs at a 60 µM aEP2 concentration. PBS-incubated NPs (empty) were used as control matching the highest NP concentration. The expression was normalized to the MC + PGE2 control moDCs. Each dot represents one donor and the bar is the mean ± SD for n = 5. *p* values were calculated on non-normalized data with one-way ANOVA with Tukey multiple comparison correction.

**Figure 3 ijms-24-01392-f003:**
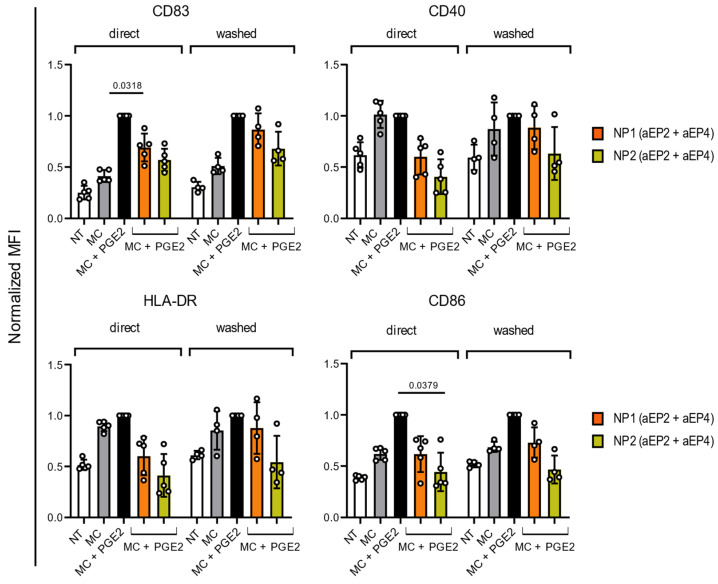
Normalized expression of CD83, CD40, HLA-DR, and CD86 on moDCs 24 h after the addition of NPs, either directly or washed cells. Controls included NT, stimulated with MC and MC with PGE2. MoDCs were stimulated with NPs directly or the cells were washed after one hour of stimulation. Two batches of NPs were used with a matched 60 µM aEP2 concentration. The expression was normalized to the MC + PGE2 control. Each dot represents one donor and the bar is the mean ±SD for n = 5 for direct and n = 4 for washed condition. *p* values were calculated on non-normalized data with one-way ANOVA with Tukey multiple comparison correction.

**Figure 4 ijms-24-01392-f004:**
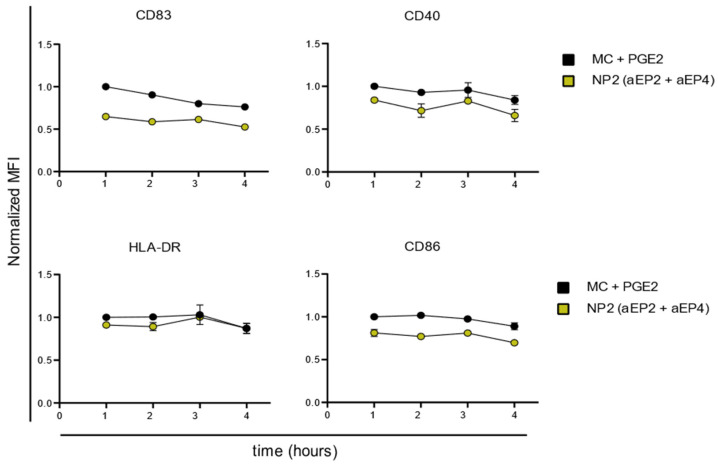
Normalized expression of CD83, CD40, HLA-DR, and CD86 on moDCs after 24 h of stimulation with NPs for varied hours. The control with MC + PGE2 was repeated for each timepoint. The expression was normalized to the MC + PGE2 expression after 1 h of stimulation. The NT and MC control are shown in Appendix A. Each dot represents the mean ± SD for n = 4.

**Figure 5 ijms-24-01392-f005:**
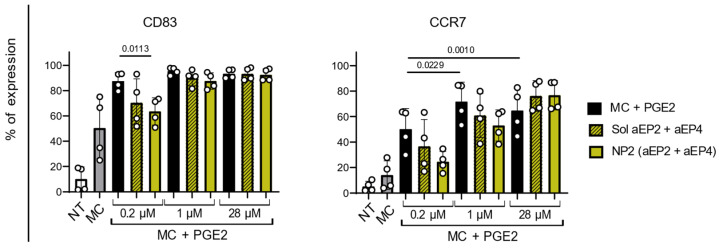
Percentage of CD83 and CCR7 expressing moDCs after 24 h of stimulation with encapsulated or soluble aEP2 + aEP4 and increasing PGE2 concentration. Control moDCs were left NT, stimulated with MC and MC with three different concentrations of PGE2: 0.2, 1, or 28 µM. MoDCs were stimulated with encapsulated or soluble aEP2 + aEP4 and MC + PGE2 with the three different PGE2 concentrations. The encapsulated and soluble antagonist concentration matched a final concentration of 60 µM aEP2 and 1.25 µM aEP4. Each dot represents one donor and the bar is the mean ± SD for n = 4. *p* values were calculated on non-normalized data with one-way ANOVA with Tukey multiple comparison correction.

**Table 1 ijms-24-01392-t001:** Characterization of nanoparticles. Size (nm) and polydispersity index (PDI) of nanoparticle (NP) formulations after lyophilization. The weight percent (wt%) of antagonist encapsulated and the encapsulation efficacy (EE%) in percent for the EP2 (aEP2) and EP4 (aEP4) antagonists (mean of n ≥ 3 ± SD).

NPs	Loading (%)	Mean Size (nm)	PDI	wt% aEP2	wt% aEP4	EE% aEP2	EE% aEP4
**NP (aEP2)**	2 wt% aEP2	226 (±11)	0.06 (±0.02)	1.24 (±1.35)	n.a. *	20.9 (±19.1)	n.a.
**NP (aEP4)**	2 wt% aEP4	244 (±33)	0.08 (±0.01)	n.a.	0.63 (±0.47)	n.a.	17.4 (±18.7)
**NP1 (aEP2 + aEP4)**	3 wt% aEP2 + 1 wt% aEP4	261 (±22)	0.13 (±0.06)	3.12 (±0.63)	0.36 (±0.28)	66.8 (±33.1)	25.0 (±25.6)
**NP2 (aEP2 + aEP4)**	3 wt% aEP2 + 2 wt% aEP4	242 (±26)	0.08 (±0.04)	3.08 (±0.74)	0.47 (±0.34)	43.6 (±13.7)	14.3 (±18.0)
**NP (empty)**	0%	211 (±8)	0.07 (±0.01)	n.a.	n.a.	n.a.	n.a.

* Not applicable (n.a.).

**Table 2 ijms-24-01392-t002:** Percentage of aEP2 and aEP4 from the initial encapsulated wt% within the NP formulations after 15 min at 37 °C (mean of n ≥ 2 ± SD).

	Antagonist in NPs (%)	
**NPs**	aEP2	aEP4
**NP (aEP2)**	0.7% (±1.0)	n.a. *
**NP (aEP4)**	n.a.	17.4% (±0.5)
**NP1 (aEP2 + aEP4)**	31.1% (±0.8)	44.0% (±1.9)
**NP2 (aEP2 + aEP4)**	32.6% (±1.3)	36.9% (±0.1)

* Not applicable (n.a.).

## Data Availability

Data are available upon reasonable request.

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
