# Peer review of "Tailored PGE2 Immunomodulation of moDCs by Nano-Encapsulated EP2/EP4 Antagonists"

_ijms, 2023, doi:10.3390/ijms24021392_

Round 1

Reviewer 1 Report

In the manuscript by Bodder et al entitled “Tailored PGE2 immunomodulation of moDCs by nano-encapsulated EP2/EP4 antagonists” the effect of PGE2R blockade on dendritic cell function is studied. This is a relevant question with respect to DC-mediated immunotherapy that is applied in the oncology field as many tumors produce PGE2 and therefore influence the effectiveness of the DC-vaccine.

The manuscript is well-written and the results are presented in clear figures.

Specific comments:

-          How well tolerated PLGA + PGE2 antagonists? Did the authors observe any cell death in the cultures? 

-          Do NP need to be taken up to be effective or not? Please speculate.

-          It is unclear what statistics have been performed. Has the statistical analysis been performed to the MC-PGE2 condition of have all conditions been compared to everything? To evaluate the effects of the incorporation of the PGE2 antagonist the authors should also statistically analyze the responses versus the empty NP. Do the PGE2 antagonist bearing NP inhibit maturation compared to empty NP? Moreover, the effect of NP (aEP2) seems to be as strong as NP (aEP2+aEP4) on DC maturation, but is not statistically significant, especially for CD40 and HLA-DR, where the strongest inhibition is observed, this seems contra-intuitive. Can the authors explain this? 

-          PGE2 not only regulated maturation of DCs but is also known as a factor that drives the Th2 promotion capacity of DCs. Is that at the same concentration as used in the maturation cocktail? And what is the role of blocking PGE2 in that respect for anti-cancer immunity. Please discuss.

-          Minor point: Description of X-axis of Figure 3 CD83 is missing as well as the p value NP empty versus NP (aEP2+aEP4).

Author Response

We thank the reviewer for their time and effort to improve the manuscript.  

  • How well tolerated PLGA + PGE2 antagonists? Did the authors observe any cell death in the cultures?  

Thank you for this comment. We now included in the results part that we did not observe differences in moDC viability with the NP concentrations used for stimulation

  •   Do NP need to be taken up to be effective or not? Please speculate. 

Based on our experiments we cannot draw firm conclusions. We now discussed in the discussion the requirement of NPs uptake for the effectiveness of the encapsulated antagonists.

We speculate that the NPs at least partially function via NP uptake due to the effectiveness of the washed NP and the stronger effect on marker expression of the NP compared to the soluble antagonists.  

  • It is unclear what statistics have been performed. Has the statistical analysis been performed to the MC-PGE2 condition of have all conditions been compared to everything? 

    To evaluate the effects of the incorporation of the PGE2 antagonist the authors should also statistically analyze the responses versus the empty NP. Do the PGE2 antagonist bearing NP inhibit maturation compared to empty NP?  

    Moreover, the effect of NP (aEP2) seems to be as strong as NP (aEP2+aEP4) on DC maturation, but is not statistically significant, especially for CD40 and HLA-DR, where the strongest inhibition is observed, this seems contra-intuitive. Can the authors explain this?  

    Thank you for pointing out this unclarity. We now compared all conditions with each other and included this in the Material and Methods section.  

    We included the significant reduction of marker expression with antagonist-encapsulated NPs compared to the empty NPs in Figure 2. For CD40 and CD86, the antagonist-loaded NPs show a significantly reduced expression compared to empty NPs. We agree that adding the significance strengthens our argument that the encapsulated antagonists specifically block MC + PGE2-induced marker expression. 

In addition, when calculating the significance of comparing all conditions with each other the NP (aEP2) also significantly downregulates CD40, HLA-DR, and CD86. Even though most of the effect seems to depend on aEP2 we would continue with the combination of both antagonists due to the lower release of antagonists and literature suggestions.  

  • PGE2 not only regulated maturation of DCs but is also known as a factor that drives the Th2 promotion capacity of DCs. Is that at the same concentration as used in the maturation cocktail? And what is the role of blocking PGE2 in that respect for anti-cancer immunity. Please discuss. 

In literature, the role of PGE2 to drive Th2 responses is still unclear, and contractionary results are published (Rubio et al. 2005; Scandella et al 2002; P KaliÅ„ski et al 1997; Lee et al 2002). It seems that different culture conditions, among others the kind of T cells (naïve), T cell/DC ratio, and other factors like pro-inflammatory cytokines lead to variations in results. In addition, we here used a significantly lower concentration of PGE2. We used 0.2 uM and previous studies between 3 and 9 uM.

Therefore, although we acknowledge the comment of the reviewer, we considered this point to be out of the scope of the article to discuss the role of blocking PGE2 for the Th2 differentiation and the impact it has on anti-cancer immunity. Of course, it is an interesting angle for further investigations. 

Rubio et al. 2005  https://doi.org/10.1093/intimm/dxh335
Scandella et al 2002 https://doi.org/10.1182/blood-2001-11-0017
P Kaliński et al 1997 https://doi.org/10.4049/jimmunol.159.1.28
Lee et al 2002 https://doi.org/10.1634/stemcells.20-5-448

  •  Minor point: Description of X-axis of Figure 3 CD83 is missing as well as the p value NP empty versus NP (aEP2+aEP4). 

Thank you for pointing this out. We corrected Figure 3 with the Axis and p value.  

Reviewer 2 Report

The authors study the potential of two E-Prostanoid (EP) receptor antagonists encapsulated in nanoparticles to immune modulate dendritic cells. These NPs should then be used to treat tumors. The study is solid and need just a few minor changes,

1.      Please clarify how Table 1 and Table2 are connected. What is the difference between wt% aEP2 (Table 1) and Antagonistic in NP (%) (Table 2). Please explain better.

2.      Please label the black bar in Figure 2-4 better. Please add what the black bar represents in the legend next to the graph where the NPs combinations are described.

Author Response

We thank the reviewers for their time and effort to improve the manuscript.

  1. Please clarify how Table 1 and Table2 are connected. What is the difference between wt% aEP2 (Table 1) and Antagonistic in NP (%) (Table 2). Please explain better.

Thank you for pointing out this unclarity. In the revised manuscript, the differences between the two tables are explained in the text.

  1. Please label the black bar in Figure 2-4 better. Please add what the black bar represents in the legend next to the graph where the NPs combinations are described

We agree with the comment. Therefore, we labeled the black bar directly in the relevant figures (Figure 2-4).

Reviewer 3 Report

The study design is appropriate because the experiments tested the role and effects of encapsulated antagonists in tumor cells.

Regarding the test results, good results were obtained. However, the discussion is somewhat lacking in describing trends in other studies and how the results obtained could be applied clinically. The reader is expected to apply the results of this study in any way. The author would like to add more about this expectation, if possible.

In figure 5, you indicated " Normalized expression of CD83, CD40, HLA-DR and CD86 on moDCs after 24 h of stim- 222 ulation with NPs for varied hours. " The experimental results did not show significant differences, but was a positive control not necessary in this test system? Supplementation of the test system being established may be necessary because there is a lack of material to determine if these results are correct. 

If you revise the manuscript, please add a supplement regarding the correctness of the experiments and the prospects for possible clinical application. I believe that their inclusion will make the paper better.

Author Response

We thank the reviewers for their time and effort to improve the manuscript.

Regarding the test results, good results were obtained. However, the discussion is somewhat lacking in describing trends in other studies and how the results obtained could be applied clinically. The reader is expected to apply the results of this study in any way. The author would like to add more about this expectation, if possible.

We added the impact of our results for the clinical application in the discussion and focused more on the outlook of the study.

In figure 5, you indicated " Normalized expression of CD83, CD40, HLA-DR and CD86 on moDCs after 24 h of stimulation with NPs for varied hours. The experimental results did not show significant differences, but was a positive control not necessary in this test system? Supplementation of the test system being established may be necessary because there is a lack of material to determine if these results are correct. 

Only the normalized expression of the maturation markers on dendritic cells after 24h was indeed shown to simplify Figure 5. In this way, the impact of the nanoparticles on the maturation of dendritic cells in response to the maturation cocktail + PGE2 could be better visualized. However, we indeed measured the non-treated and maturation cocktail controls which are shown in Supplementary Figure 6. To clarify this, we adjusted the Legend of Figure 5: NT and MC control were measured for the experiment and are depicted in Supplementary Figure 

If you revise the manuscript, please add a supplement regarding the correctness of the experiments and the prospects for possible clinical application. I believe that their inclusion will make the paper better.